# Gut Microbiota Associated with Clinical Relapse in Patients with Quiescent Ulcerative Colitis

**DOI:** 10.3390/microorganisms10051044

**Published:** 2022-05-18

**Authors:** Hiroaki Kitae, Tomohisa Takagi, Yuji Naito, Ryo Inoue, Yuka Azuma, Takashi Torii, Katsura Mizushima, Toshifumi Doi, Ken Inoue, Osamu Dohi, Naohisa Yoshida, Kazuhiro Kamada, Kazuhiko Uchiyama, Takeshi Ishikawa, Hideyuki Konishi, Yoshito Itoh

**Affiliations:** 1Department of Molecular Gastroenterology and Hepatology, Graduate School of Medical Science, Kyoto Prefectural University of Medicine, 465 Kawaramachi Hirokoji, Kamigyo-ku, Kyoto 602-8566, Japan; hkitae@koto.kpu-m.ac.jp (H.K.); ayuka@koto.kpu-m.ac.jp (Y.A.); t-taka@koto.kpu-m.ac.jp (T.T.); mizusima@koto.kpu-m.ac.jp (K.M.); t-doi@koto.kpu-m.ac.jp (T.D.); keninoue71@koto.kpu-m.ac.jp (K.I.); osamu-d@koto.kpu-m.ac.jp (O.D.); naohisa@koto.kpu-m.ac.jp (N.Y.); k-kamada@koto.kpu-m.ac.jp (K.K.); k-uchi@koto.kpu-m.ac.jp (K.U.); iskw-t@koto.kpu-m.ac.jp (T.I.); hkonishi@koto.kpu-m.ac.jp (H.K.); yitoh@koto.kpu-m.ac.jp (Y.I.); 2Department for Medical Innovation and Translational Medical Science, Kyoto Prefectural University of Medicine, Kyoto 602-8566, Japan; 3Department of Human Immunology and Nutrition Science, Kyoto Prefectural University of Medicine, Kyoto 602-8566, Japan; ynaito@koto.kpu-m.ac.jp; 4Laboratory of Animal Science, Department of Applied Biological Sciences, Faculty of Agriculture, Setsunan University, Osaka 573-0101, Japan; ryo.inoue@setsunan.ac.jp

**Keywords:** quiescent ulcerative colitis, fecal microbiota, ulcerative colitis relapse, LEfSe

## Abstract

The microbiota associated with relapse in patients with quiescent ulcerative colitis (qUC) remains unclear. Our objective was to analyze the fecal microbiota of Japanese patients with qUC and identify the relapse-associated microbiota. In this study, 59 patients with qUC and 59 healthy controls (HCs) were enrolled (UMIN 000019486), and their fecal microbiota was compared using 16S rRNA gene amplicon sequencing. We followed their clinical course up to 3.5 years and analyzed the relapse-associated microbiota. Potential functional changes in the fecal microbiota were evaluated using PICRUSt software and the Kyoto Encyclopedia of Genes and Genomes database. There were significant differences in fecal microbiota diversity between HC and qUC subjects, with 13 taxa characterizing each subject. Despite no significant difference in variation of microbiota in a single sample (α diversity) between patients in sustained remission and relapsed patients, the variation in microbial communities between samples (β diversity) was significantly different. *Prevotella* was more abundant in the sustained remission patients, whereas *Faecalibacterium* and *Bifidobacterium* were more abundant in the relapsed patients. We clustered the entire cohort into four clusters, and Kaplan–Meier analysis revealed the subsequent clinical course of each cluster was different. We identified 48 metabolic pathways associated with each cluster using linear discriminant analysis effect size. We confirmed the difference in microbiota between patients with qUC and HCs and identified three genera associated with relapse. We found that the clusters based on these genera had different subsequent clinical courses and activated different metabolic pathways.

## 1. Introduction

The intestinal microbiota is known to be involved in the pathogenesis of various diseases, ranging from local gastroenterological disorders to neurological, respiratory, metabolic, hepatic, and cardiovascular diseases [1,2]. It has previously been reported that the intestinal microbiota differs between healthy individuals and those with Inflammatory bowel diseases (IBDs), including ulcerative colitis (UC) and Crohn’s disease [3,4]. A reduction in the diversity and a low abundance of Firmicutes are the definitive changes observed in the microbiota of patients with IBD [5,6]. Moreover, there are conflicting reports [6,7,8,9,10,11,12,13,14] regarding the presence of Bacteroidetes phyla in IBD, with Imhann et al. reporting an increase in IBD, while Santoru et al. reporting a decrease in IBD. 

A recent review reported decreased *Akkermansia* and *Eubacterium rectale* in patients with UC [15]. A previous study of the gut microbiota in patients with active UC showed that *Bifidobacterium* and *Lactobacillus* increased, and *Faecalibacterium* decreased in colon biopsy specimens [16]. A meta-analysis of UC patients showed that patients with an active disease episode had a lower abundance of *Clostridium coccoides*, *Clostridium leptum*, *Faecalibacterium prausnitzii*, and *Bifidobacterium* than patients with quiescent UC (qUC) [17]. Although the differences in the gut bacterial composition among healthy individuals and patients with inactive and active IBD have widely been reported, the results of such studies have been controversial. These specific changes in the gut microbiota are thought to be involved in IBD pathogenesis [18], but few have been considered definitive at this time. 

Additionally, to our knowledge, there are no studies to date on fecal microbiota involved in the clinical relapse in patients with qUC. Therefore, in this study, we collected fecal samples from UC patients in clinical remission, followed their clinical course for 3.5 years after fecal sampling, and analyzed the fecal microbiota associated with UC relapse.

## 2. Materials and Methods

### 2.1. Ethics

This study was approved by the ethics committee of the Kyoto Prefectural University of Medicine (ERB-C-534) and was registered at the University Hospital Medical Information Network Center (UMIN 000019486). Patient care was managed at the Division of Gastroenterology, Kyoto Prefectural University of Medicine. Written informed consent was obtained from all study participants before their enrollment. The study was conducted in accordance with the Declaration of Helsinki. 

### 2.2. Patients

A total of 118 subjects, comprising 59 patients with UC and 59 HCs, were enrolled. We recruited consecutive patients with quiescent UC in our outpatient clinic from November 2016 to September 2017. Clinical remission was defined as a Lichtiger index score of ≤4 [19] and no history of antibiotic use within three months. A total of 59 age- and sex-matched volunteers were selected from our previous study [2] as HCs (Table 1). We requested a specimen match without giving any clinical information other than age and sex to a third party who was unaware of the purpose of this study. These volunteers were in good health and had no evidence of significant gastrointestinal inflammatory diseases including IBD, functional gastrointestinal disorders such as irritable bowel syndrome, or infectious enteritis within the past three months. These volunteers had not taken any prebiotics or probiotics within the past three months. Additional exclusion criteria included a history of underlying malignant disease, administration of antibiotics, and gastric acid secretion inhibitors within the past three months. In addition, subjects with serious metabolic, respiratory, cardiologic, renal, hepatic, hematologic, neurologic, or psychiatric functions were excluded. Pregnant or lactating individuals were also excluded.

The 59 patients with qUC were followed up for 3.5 years, and the fecal microbiota of patients with relapse was compared with that of patients in sustained remission. Clinical relapse was defined as clinical or endoscopic deterioration requiring therapeutic modification. Comparisons of background data between groups were analyzed using Pearson’s chi-squared test for categorical variables and Welch’s *t*-test for continuous variables.

### 2.3. Sample Collection and DNA Extraction

Fecal samples were obtained, and fecal bacterial composition analysis was performed [20,21,22]. Fecal samples (5 mm × 2 mm) were obtained using guanidine thiocyanate solution (Feces Collection Kit; Techno Suruga Lab, Shizuoka, Japan). The fecal samples were agitated vigorously and stored at 25 °C or lower for up to 7 days until DNA extraction.

Genomic DNA was isolated using the NucleoSpin Microbial DNA Kit (Macherey–Nagel, Duren, Germany). Approximately 500 µL of the stored fecal sample was placed in a microcentrifuge tube with 100 µL of elution buffer (BE). The mixture was then placed into a NucleoSpin^®^ bead tube with proteinase K and mechanically bead-beaten for 12 min at 30 Hz in the TissueLyser solution. The subsequent extraction procedure was performed according to the manufacturer’s instructions. Extracted DNA samples were purified using Agencourt AMPure XP (Beckman Coulter, Brea, CA, USA). 

### 2.4. Sequencing of 16S rRNA Gene

A two-step polymerase chain reaction (PCR) was performed on the purified DNA samples to obtain sequence libraries. The first PCR was performed to amplify, and a 16S (V3–V4) metagenomic library construction kit for NGS (Takara Bio Inc., Kusatsu, Japan) was used with primer pairs 341F (5′- TCGTCGGCAGCGTCAGATGTGTATAAGAGACAGCCTACGGGNGGCWGCAG-3′) and 806R (5′-GTCTCGTGGGCTCGGAGATGTGTATAAGAGACAGGGACTACHVGGGTWTCTAAT-3′) corresponding to the V3–V4 region of the 16S rRNA gene. The second PCR was performed to add the index sequences for the Illumina sequencer with a barcode sequence using the Nextera XT Index Kit (Illumina, San Diego, CA, USA). The prepared libraries were subjected to sequencing of 250 paired-end bases using the MiSeq Reagent v2 Kit and MiSeq (Illumina) at the Biomedical Center of Takara Bio, Japan.

### 2.5. Microbiome Analysis

The processing of sequence data, including chimera check, ASV definition, and taxonomy assignment, was performed using the Quantitative Insights Into Microbial Ecology 2 (QIIME2) version 2020.8 [22]. Singletons were excluded in this study. The taxonomy assignment of the resulting ASV was completed using the Sklearn classifier algorithm against the Greengenes database version 13_8 (99% OTU dataset). 

The Chao1 (ASV richness estimation) and Shannon (ASV evenness estimation) phylogenetic diversity indices were calculated for variation in microbiota in a single sample (α diversity) and were statistically analyzed using Wilcoxon’s rank-sum test. The variation in microbial communities between samples (β diversity) was estimated using the UniFrac metric [23], which calculates the distances between the samples, visualized via PCoA, and was statistically examined using permutational multivariate analysis of variance (PERMANOVA). The unweighted UniFrac distance was calculated purely based on sequence distances and did not include abundance information. By contrast, the weighted UniFrac distance considered relative abundances and included both sequence and abundance information. The final figures were generated using the software QIIME2 (ver. 2020.8). Receiver operating characteristic (ROC) analysis based on logistic regression analysis was performed with JMP Pro 14.0.0 (SAS Institute Inc., Cary, NC, USA). Areas under the curve (AUC) values were calculated from the ROC curve as an indicator of the predictive value. Hierarchical clustering analysis was used in this study as an exploratory procedure using the Ward method with JMP.

### 2.6. Predictive Functional Profiling of Gut Microbial Communities

The software Phylogenetic Investigation of Communities by Reconstruction of Unobserved States (PICRUSt) v2.1.4 [24] was used to gain more insight into the metagenomic-based function of the microbiome in each cluster. PICRUSt was used to obtain relative Kyoto Encyclopedia of Genes and Genomes (KEGG) pathway abundance information derived from metagenomic data. The predicted data were collapsed into hierarchical categories (KEGG Level 1, 2, and 3), and the relative abundances of the gut metabolic functions were calculated. We calculated the nearest sequenced taxon index and excluded five ASVs with the nearest sequenced taxon index > 2. We focused on levels 1, 2, and 3 to investigate the differences in metabolic pathways between the clusters. 

### 2.7. Linear Discriminant Analysis Effect Size (LEfSe) Analysis

Linear discriminant analysis effect size (LEfSe) [25], a method for biomarker discovery, was used as a part of the Galaxy application (http://huttenhower.org/galaxy, accessed on 1 December 2020) to determine gut bacteria and their metabolic pathways that best characterized each study group. The linear discriminant analysis score (LDA score) indicates the effect size of each pathway. Taxa with LDA scores > 4, *p* < 0.05 for the microbiota and LDA scores > 2, *p* < 0.05 for the KEGG metabolic pathway were considered significant.

## 3. Results

### 3.1. Comparing the Composition of Gut Microbiota between Patients with qUC and HCs

In total, 4585 amplicon sequence variants (ASVs) were detected from 5,975,710 high-quality sequence reads (average: 50,641.6 reads/sample). The rarefaction curve reached an apparent plateau at 3000 reads, suggesting that most of the sample variation was covered. First, we evaluated the diversity of the gut microbiota between qUC and HCs. The principal coordinate analysis (PCoA) plot based on weighted and unweighted UniFrac distances was calculated as an index of β diversity. While the unweighted UniFrac did not consider relative abundance when comparing communities and considered only the overlap of genera, the weighted UniFrac considered the relative abundance of each genus. PCoA revealed that the overall composition of the gut microbiome for HC and UC samples was significantly different in unweighted and weighted distances (*p* < 0.01). Regarding α diversity, Chao1 (richness), observed species (richness), and Shannon (evenness) indices in qUC fecal samples were significantly lower than those in the HC samples (Figure 1a–e). Table 2 shows the 10 most abundant genera in HCs and patients with qUC colitis, with 7 of the 10 genera being the same.

LEfSe analysis revealed seven taxa characterizing UC, and six taxa characterizing HC samples (Figure 1f). At the phylum level, Actinobacteria characterized UC and Bacteroidetes characterized HC. The mean abundance of Actinobacteria was 8.02% in HCs and 18.6% in patients with UC, while the mean abundance of Bacteroidetes was 26.09% in HCs and 20.52% in patients with UC. At the genus level, *Bifidobacterium* characterized UC samples. Following a previous report [3,4], the abundance of *Clostridium* and *Faecalibacterium* was low, although they were not significantly different within the groups (data not shown).

### 3.2. Fecal Microbiota Associated with Relapse in qUC Patients

Of the 59 UC patients observed for a long-term of 3.5 years, 19 patients (32.2%) had clinically relapsed (Figure 2a). This relapse rate is comparable with that previously reported in patients with qUC, which was 26.3–37.3% annually [26,27]. Regarding sex, age, body mass index, UC disease duration, observation period until relapse, medication, and intake of probiotics, there were no significant differences in patient backgrounds between patients with sustained remission (SusRem) and relapse patients (Table 3). UC with proctitis type was observed only in the SusRem group. 

All α-diversity indices, including Chao1, observed ASVs, and Shannon indices, showed no significant differences between SusRem and relapse groups. The PCoA plot of weighted UniFrac distance indicated that the fecal microbiota differed significantly between SusRem and relapse groups (*p* = 0.04). However, the PCoA plot of unweighted UniFrac distance showed no significant difference between those groups (Figure 2b–f). These suggest that there were differences in the composition between each group. LEfSe analysis revealed two taxa characterizing the SusRem group, and six taxa characterizing the relapse group (Figure 2g). At the phylum level, Actinobacteria characterized the relapse group, and the mean abundance of Actinobacteria was 25.19% in the relapse group and 15.46% SusRem group. At the genus level, *Prevotella* characterized SusRem samples, and *Faecalibacterium* and *Bifidobacterium* characterized relapse samples. The mean abundance of *Prevotella* was 0.09% in the relapse group and 3.01% in the SusRem group, the mean abundance of *Faecalibacterium* was 8.92% in the relapse group and 5.47% in the SusRem group, and the mean abundance of *Bifidobacterium* was 21.34% in the relapse group and 11.73% in the SusRem group. In addition, the abundance of the three genera was calculated by the presence or absence of probiotic intake. In the group of patients taking probiotics, the mean abundance of *Prevotella* was 0.15% in the relapse group and 3.45% in the SusRem group, and the mean abundance of *Faecalibacterium* was 9.45% in the relapse group and 3.46% in SusRem group; the mean abundance of *Bifidobacterium* was 18.47% in the relapse group and 11.01% in the SusRem group. In the group of patients not taking probiotics, the mean abundance of *Prevotella* was 0% in the relapse group and 3.78% in the SusRem group, the mean abundance of *Faecalibacterium* was 8.20% in the relapse group and 7.32% in the SusRem group, and the mean abundance of *Bifidobacterium* was 25.29% in the relapse group and 12.88% in the SusRem group. 

In multiple logistic regression models based on the abundance of three genera—*Prevotella*, *Faecalibacterium*, and *Bifidobacterium*—the probability of relapse was obtained as follows:$$\begin{array}{cc} Probability=1 & /\{1+exp\;(67.4609012869072\times Prevotella\\ & -\;18.0001654746467\times Faecalibacterium\\ & -\;7.72194050290922\times Bifidobacterium\\ & +2.91205879564624)\} \end{array}$$

Then, we calculated the ROC curves and AUCs based on multivariate logistic regression analysis with these three genera and based on univariate logistic regression analysis for each genus (Figure 3a). The AUC for the combination of the three genera was 0.856, which was significantly higher than the AUCs based on each genus. Therefore, the prediction of UC relapse using a combination of these three genera was valid. Further analysis of 49 UC patients, excluding 10 patients with proctitis, showed that the abundance of *Prevotella* was significantly lower, and the abundance of *Faecalibacterium* and *Bifidobacterium* was significantly higher in the 19 relapse patients than those in 30 SusRem patients, which was similar to the results obtained in a former analysis including proctitis (data not shown). 

### 3.3. Pathway Analysis in Identified Clusters

Based on the three genera—*Prevotella*, *Faecalibacterium*, and *Bifidobacterium*—a clustering analysis was performed for the entire cohort, including patients with qUC and HCs. They were classified into four clusters using Ward’s hierarchical clustering method (Figure 3b). The *Prevotella*-rich cluster had 12 individuals (HCs = 8; UC in sustained remission = 4). The *Faecalibacterium*-rich cluster had 23 individuals (HCs = 17; UC = 6 (3 relapsed; 3 sustained remission)). The *Bifidobacterium*-rich cluster had 22 individuals (HCs = 3; UC = 19 (10 relapsed, 9 sustained remission)). The fourth (other) cluster had 61 subjects (HCs = 31, UC = 30 (6 relapsed, 24 sustained remission)). Table 4 shows the 10 most common genera for each cluster. As expected, in each cluster characterized by the three identified genera, each genus occupies a high abundance of each cluster. On the other hand, the genera composing the other cluster are not particularly increasing in the other genera that were not focused on in this study. The Kaplan–Meier curve showed that the clinical course of each cluster was significantly different according to the log-rank test (Figure 3c). The relapse time was different in each cluster; the median (minimum–maximum) time to relapse was 357 (342–392) days in the *Faecalibacterium*-rich cluster and 302 (27–1065) days in the *Bifidobacterium*-rich cluster, with no relapse in the *Prevotella*-rich cluster.

Subsequently, based on the identified KEGG pathways, LEfSe analysis revealed 48 pathways characterizing each cluster (Figure 4). In the hierarchical category of KEGG-Level-2, “cell growth and death”, transcription, “metabolism of cofactors and vitamins”, and “metabolism of the terpenoids and polyketides” pathways were activated in the *Prevotella*-rich cluster. Furthermore, “cell motility”, “lipid metabolism”, and “immune system” pathways were activated in the *Faecalibacterium*-rich cluster. “Transport and catabolism”, “infectious disease, parasitic”, and “xenobiotics, biodegradation and metabolism” pathways were activated in the *Bifidobacterium*-rich cluster.

## 4. Discussion

We confirmed in a single-center, prospective, observational study of fecal microbiota in patients with qUC and their clinical course that the microbiota of HCs and patients with qUC were different, although their clinical symptoms were the same as previously reported [3,4]. We investigated the microbiota in patients with qUC with clinical relapse and found *Prevotella*, *Faecalibacterium*, and *Bifidobacterium* as the gut microbial genera associated with relapse. To the best of our knowledge, this study is the first report of microbiota related to relapse within 3.5 years of contracting qUC. It may help us understand the pathogenesis of IBD and identify the predictors of relapse.

In the present study, approximately 30% of patients with UC in clinical remission relapsed, consistent with that in previous reports [26,27]. It has been reported that the alpha diversity of the microbiota in biopsy specimens from the relapse group was lower than that from the non-relapse group, and this study showed a similar trend but no significant difference [28]. PCoA based on the weighted UniFrac distance showed a difference between patients in relapse and SusRem groups, indicating that the composition of the fecal microbiota was different between patients with relapse and sustained remission. The phylum Actinobacteria was found to be more abundant in the relapse group than that in the SusRem group, reflecting a rich abundance of *Bifidobacterium*. The most striking finding of this study is that the genus *Prevotella* was abundant in the SusRem group, while *Faecalibacterium* and *Bifidobacterium* were abundant in the relapse group. When analyzed by the presence or absence of probiotics, the tendency of the results for these three genera was similar. 

The genus *Prevotella* is known to contain a set of bacterial genes for the hydrolysis of cellulose and xylan. A positive correlation between the abundance of *Prevotella* and short-chain fatty acids (SFCA) was reported in rural African children who were found to consume a more fiber-rich diet than European children [29]. In the present study, the *Prevotella*-rich cluster included 12 subjects with no UC relapse and was characterized by the activation of 20 pathways (Figure 4a), including the riboflavin metabolism pathway. Riboflavin, also known as vitamin B2, has antioxidant, anti-inflammatory, and microbiome-modulatory properties. Supplementation of riboflavin in Crohn’s disease patients resulted in reduced clinical symptoms and systemic oxidative stress and mixed anti-inflammatory effects [30]. In addition, riboflavin has been reported to be abundant in healthy diets [31]. The *Prevotella*-rich clusters predominantly included HCs and did not include patients with UC relapse, suggesting a link to the anti-inflammatory effects of riboflavin. The RNA polymerase pathway, which had the highest LDA score at KEGG Level-3 in this cluster, is a pathway involved in RNA transcription, including microRNAs that have been implicated in immune regulation [32]. As its activation has a wide range of implications, it is difficult to understand its meanings. The Lipopolysaccharide biosynthesis pathway has been suggested to be involved in the anti-inflammation of CD [33], and we think that this might also be true for UC in this study. We consider that the intestinal environment may be involved in reducing inflammation in *Prevotella*-rich clusters.

A meta-analysis study reported a lower abundance of *Faecalibacterium* in patients with active UC than in patients with qUC or healthy individuals, indicating that *Faecalibacterium* may play a crucial role in inhibiting intestinal inflammation [34]. In the present study, *Faecalibacterium* was less abundant in patients with qUC than that in HCs; however, it was significantly more abundant in the relapse group than that in the SusRem group. *Faecalibacterium prausnitzii* is one of the main butyrate producers found in the intestine [35,36]. Butyrate is a short-chain fatty acid (SCFA) produced by gut bacteria utilizing dietary fiber and is an important intestinal epithelial nutrient and a mucosal immune system regulator [37]. *Faecalibacterium*-rich clusters were characterized by 11 pathways (Figure 4a), including the lipid metabolism and fatty acid biosynthesis pathways. In a previous report, the abundance of *Faecalibacterium* was associated with the activation of fatty acid biosynthesis pathways [38], consistent with the results of the present study. In the *Faecalibacterium*-rich cluster, the activation of fatty acid biosynthesis may regulate short-term inflammation in UC with increased SCFA production. Notably, more than 70% of subjects in the *Faecalibacterium*-rich cluster were HCs, suggesting that the *Faecalibacterium*-rich gut environment may not be directly associated with UC deterioration. Notably, more than 70% of subjects in the *Faecalibacterium*-rich cluster were HCs, suggesting that the *Faecalibacterium*-rich gut environment may not be directly associated with UC deterioration. *Faecalibacterium* varies in butyrate production among its strains, and an association between strain composition in the gut microbiota and butyrate concentration in feces has been reported [39]. In *Faecalibacterium*-rich clusters, the composition of *Faecalibacterium* strains may differ between HCs and patients with UC. In our study, *Faecalibacterium* was associated with UC relapse, while atopic dermatitis symptoms and *Faecalibacterium* have been reported to be positively correlated [40].

A recent review reported that the abundance of the *Bifidobacterium* genus is controversial in patients with UC [15]. In this study, we found that the abundance of *Bifidobacterium* was higher in patients with UC than that in HCs, and among patients with UC, it was higher in the relapse group than that in the SusRem group. Moreover, in the *Bifidobacterium*-rich cluster, approximately half of the patients with UC relapsed, as in the *Faecalibacterium*-rich cluster; however, more than 85% of the entire cluster included patients with UC, compared with less than 30% of that in the *Faecalibacterium*-rich cluster. These results may indicate that the intestinal environment related to UC relapse is considerably different between *Faecalibacterium*-rich and *Bifidobacterium*-rich clusters. Previously, *Bifidobacterium* and *Lactobacillus* were reported to increase in Chinese patients with active UC [16]; this report was consistent with the results of the current study, which showed early relapse of UC in the *Bifidobacterium*-rich cluster (Figure 3c). The abundance of *Bifidobacterium adrescentis* and *Bifidobacterium longum* are known to decrease with the consumption of low fermentable oligosaccharides, disaccharides, monosaccharides, and polyols (FODMAP) diet, which has been reported to reduce gut symptoms in quiescent IBD [41]. *Bifidobacterium longum* diverted immune responses toward a proinflammatory or regulatory profile, suggesting that it could drive immune responses in different directions [42]. Here, the abundance of *B. adrescentis* and *B. longum* in the *Bifidobacterium*-rich cluster was greater than that in all other clusters (data not shown). Although oral administration of some *Bifidobacterium* strains has been beneficial for UC [43], considering the results of the present study and previous reports, a high abundance of *Bifidobacterium* in the gut itself may not be a beneficial intestinal environment for UC. The primary and secondary bile acid biosynthesis pathways are activated in the *Bifidobacterium*-rich cluster leading to a probable high bile acid concentration in the gut. Secondary bile acids are known to be produced from primary bile acids by the action of the human gut microbiota [44], become relatively hydrophobic due to 7α-dehydroxylation, and have strong cytotoxicity [45]. *Bifidobacterium* is known to accumulate primary unconjugated bile acids and probably secondary unconjugated bile acids in its cytoplasm [46] and may have implications in decreasing chronic inflammation through this bile acid accumulation mechanism [47].

Interestingly, the *Bifidobacterium*-rich cluster had a low abundance of *Faecalibacterium* (Figure 3b). It has been shown that *F. prausnitzii* is also highly sensitive to a slight increase in physiological concentrations of bile salts, as its growth is compromised by bile salt concentrations of 0.5% (*w*/*v*) [48]. These facts led us to consider the possibility that in *Bifidobacterium*-rich clusters, the activation of bile acid biosynthesis may increase cytotoxic secondary bile acids, and the inhibition of *Faecalibacterium* growth may decrease SCFAs, thus creating an inflammation-prone intestinal environment. 

In this study, we focused on *Prevotella*, *Faecalibacterium*, and *Bifidobacterium* and found a trend between gut microbiota and relapse in UC patients in remission. In the group with high *Prevotella* abundance, two-thirds of the subjects were HCs, and the remaining patients with UC showed no relapse. In the *Faecalibacterium*-rich group, more than 70% of the subjects were HCs, and half of the remaining UC patients had relapsed. All the relapses occurred after more than 300 days, and it is unlikely that the gut microbiota at the time of fecal collection reflects the relapsing fecal microbiota. In contrast, less than 15% of the *Bifidobacterium*-rich group were HCs, and more than half of the remaining patients with UC had relapsed, often at a relatively early stage. As described above, *Prevotella* is associated with a high-fiber diet, and *Faecalibacterium* and *Bifidobacterium* may also interact with and influence the intestinal environment. The clusters characterized by each genus were also found to differ in the proportions of HCs and patients with UC, as well as in the subsequent clinical course of patients with UC alone. We hypothesized that artificially altering the intestinal microbiota by diet or fecal transplantation during the remission period may alter the subsequent clinical course in patients with UC. 

About half of all specimens belonged to the fourth (Other) cluster (Figure 3c), which is difficult to characterize and has a relatively low occupancy of the three genera of interest in this study. They were characterized by the low abundance of the three genera, and other factors may have influenced the clinical conditions; therefore, further investigation is necessary to characterize and understand the fourth cluster in detail. 

This study has several limitations. First, racial differences are known to influence the microbiome [49] and IBD [50]. It remains unclear whether the findings of this study are universal because this study was conducted at a single institution in Japan, and the number of subjects was limited. Second, the microbial composition is known to be markedly different between fecal and mucosal samples [51]. Thus, the fecal microbiota does not reflect the condition of the intestinal mucosal surface [52]. Investigation of the mucosa-associated microbiota may, therefore, be the next step in understanding qUC. Third, we did not perform metabolite analyses, and they may not match the metabolic pathways that were inferred to be activated. Therefore, the predicted metabolic pathway analysis in this study needs to be considered as a draft, and confirmatory studies using metabolite analysis may provide useful insights into the functional aspects of the various clusters and abundant bacterial genera in the pathogenesis of UC.

## 5. Conclusions

We evaluated the difference in the microbiota between qUC and HC and subsequently identified the fecal microbiota associated with relapse. By clustering analysis of the three genera associated with relapse—*Prevotella*, *Faecalibacterium*, and *Bifidobacterium*—we were able to divide the samples into groups with different proportions of patients with UC and different metabolic pathways assumed to be activated. Although larger, long-term studies are needed in the future, the analysis of fecal samples from patients with UC in remission could indicate the potential for subsequent relapse and whether dietary and treatment modifications are required in clinical remission.

## Figures and Tables

**Figure 1 microorganisms-10-01044-f001:**
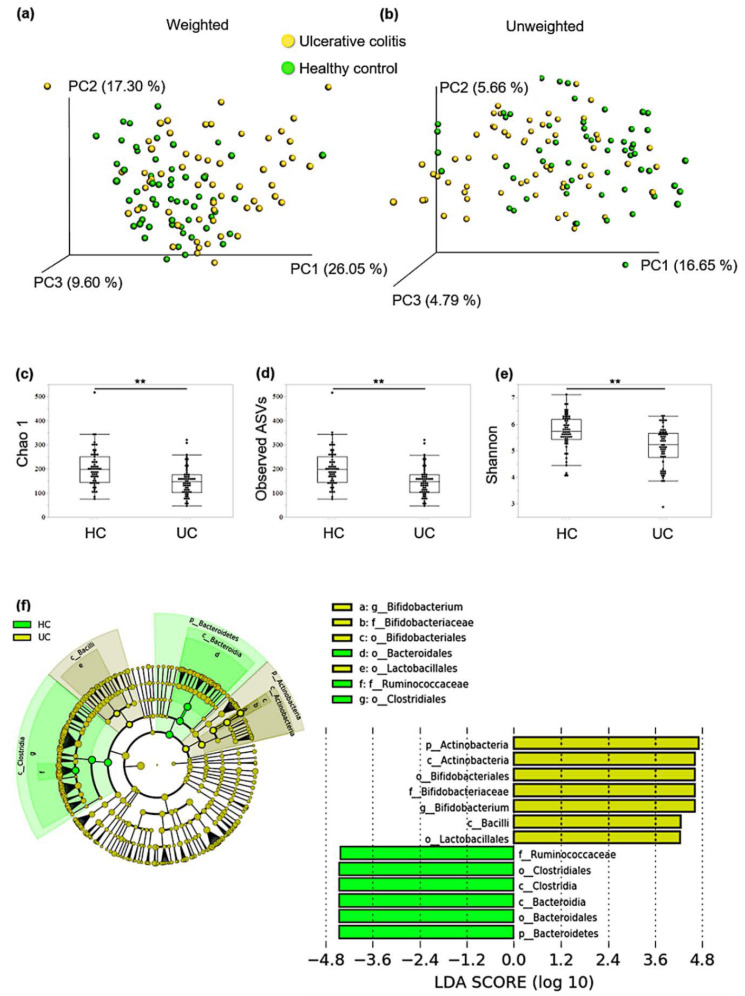
Comparison of the structure of gut microbiota between healthy controls (HCs) and patients with ulcerative colitis (UC). Principal coordinate analysis (PCoA) plots of gut microbiota based on (**a**) weighted and (**b**) unweighted UniFrac distances as β-diversity was calculated (PERMANOVA, *p* < 0.01). Yellow dots indicate patients with UC and green dots indicate HCs; As for α-diversity, (**c**) Chao 1 (ASV richness estimation); (**d**) observed ASVs; (**e**) Shannon (ASV evenness estimation) indices were significantly lower in patients with UC than those in HCs; (**f**) linear discriminant analysis effect size (LEfSe) analysis identified taxa that characterized each group. Cladogram of LEfSe analysis results. Yellow-shaded areas indicate taxa that characterize UC, and green-shaded areas indicate taxa that characterize HCs. Linear discriminant analysis (LDA) scores for identified taxa of patient of UCs (7 taxa) and HC (6 taxa) was shown. ** *p* < 0.01.

**Figure 2 microorganisms-10-01044-f002:**
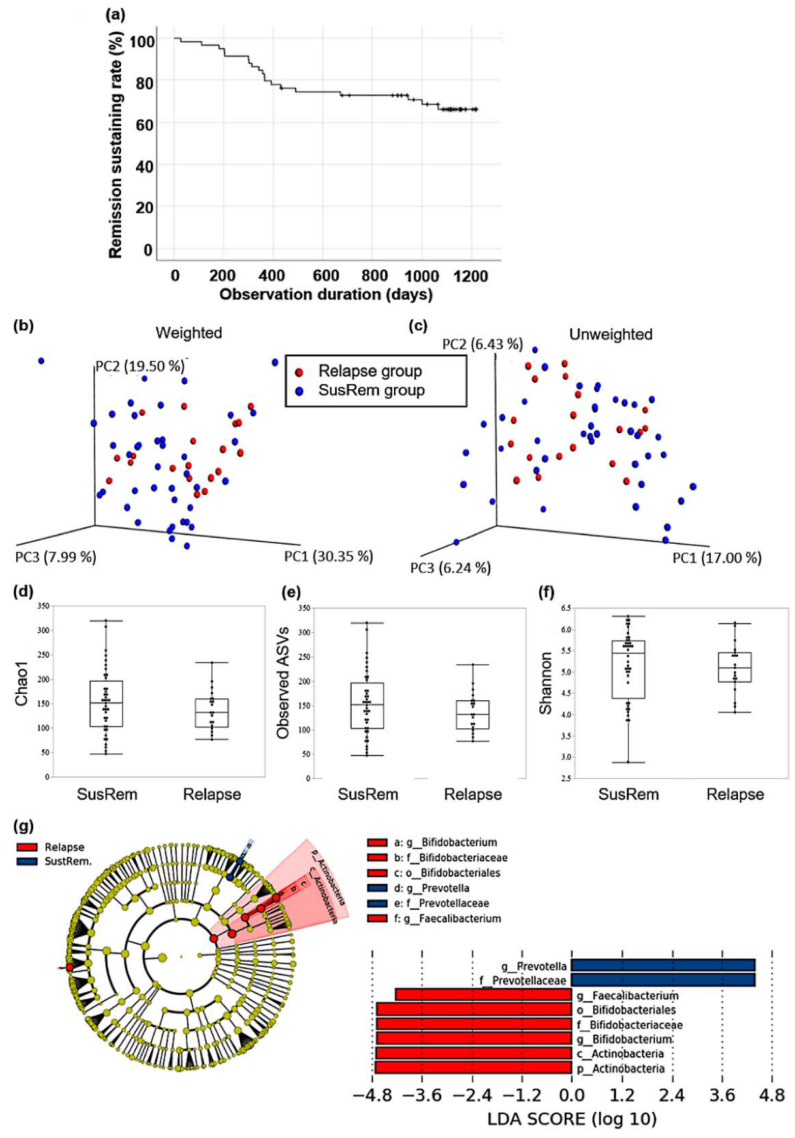
Comparison of the structure of gut microbiota between quiescent ulcerative colitis patients who sustained remission and relapsed: (**a**) Kaplan–Meier curve for sustained remission showed about 30% relapse in up to 3.5 years in the clinical course of quiescent ulcerative colitis patients. Principal coordinate analysis (PCoA) plots of gut microbiota based on (**b**) weighted and (**c**) unweighted UniFrac distances; (**d**) Chao 1 (ASV richness estimation); (**e**) observed ASVs; (**f**) Shannon (ASV evenness estimation) indices as α-diversity indices did not differ between sustained remission (SusRem) and relapsed subjects (Relapse); (**g**) linear discriminant analysis effect size (LEfSe) analysis identified taxa that characterized each group. Cladogram of LEfSe analysis results. Red-shaded areas indicate taxa that characterize relapse group, and blue-shaded areas indicate taxa that characterize SusRem group. Linear discriminant analysis (LDA) scores for identified taxa of SusRem (2 taxa) and Relapse (6 taxa) are shown.

**Figure 3 microorganisms-10-01044-f003:**
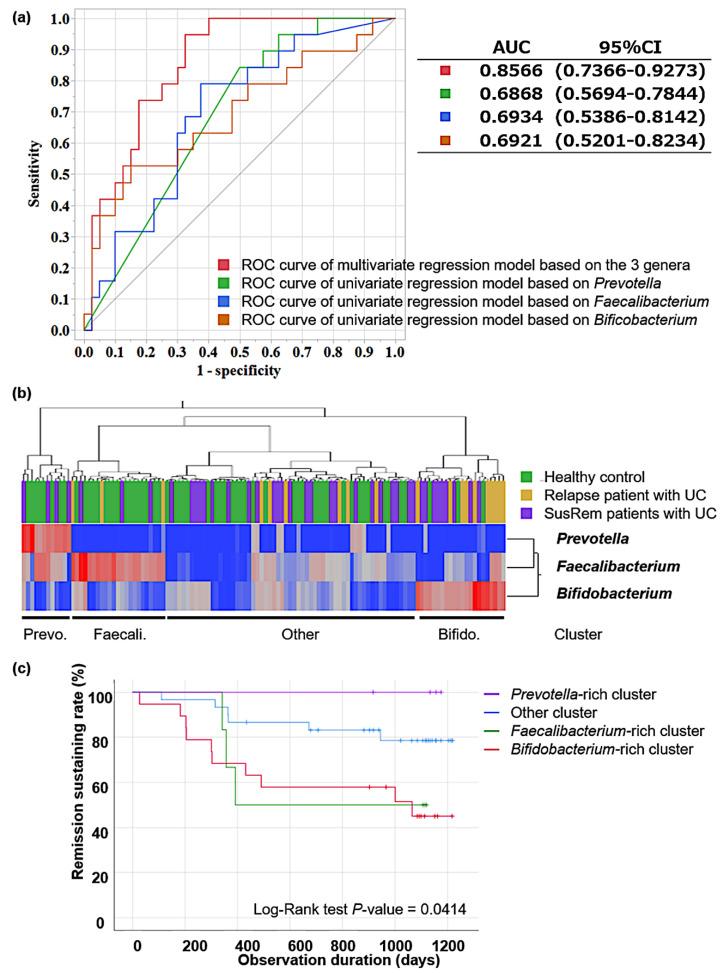
Comparison of predictors of ulcerative colitis relapse and clustering analysis based on the three genera, and their prognosis for relapse: (**a**) the receiver operating characteristics (ROC) curves based on multivariate logistic regression analysis with the abundance of *Prevotella*, *Faecalibacterium*, *Bifidobacterium* and based on univariate logistic regression analysis with each genus; (**b**) clustering analysis for the whole cohort, including quiescent ulcerative colitis patients and healthy controls via Ward’s hierarchical method based on the abundance of *Prevotella*, *Faecalibacterium,* and *Bifidobacterium.* In the upper section, green indicates healthy subjects, yellow indicates relapsed patients with ulcerative colitis, and purple indicates SusRem patients with ulcerative colitis. In the lower section, the abundance of each genus is indicated by color change, with red indicating high abundance and blue low abundance; (**c**) Kaplan–Meier analysis according to four clusters (*Prevotella*-rich, *Faecalibacterium*-rich, *Bifidobacterium*-rich, and other clusters) and log-rank test showing differences in the clinical course of each cluster. SusRem, sustained remission; Prevo., *Prevotella*; Faecali., *Faecalibacterium*; Bifido., *Bifidobacterium*.

**Figure 4 microorganisms-10-01044-f004:**
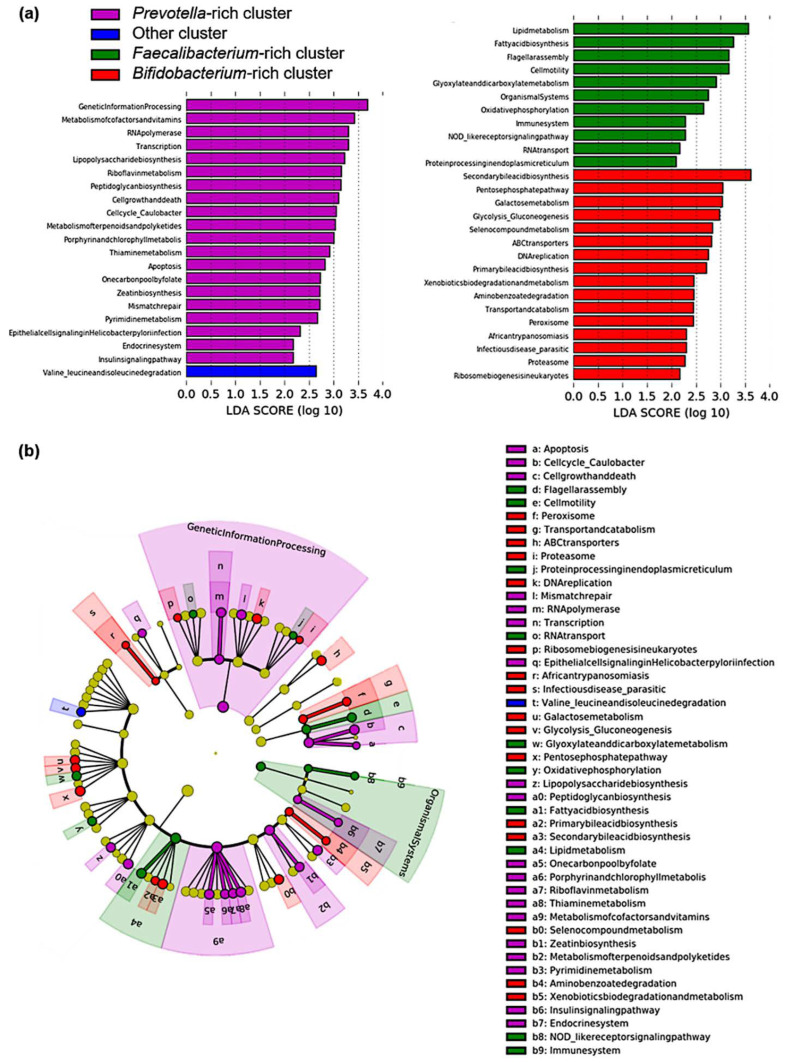
Pathway analysis based on identified clusters. Linear discriminant analysis effect size (LEfSe) analysis identified the assumed activated Kyoto Encyclopedia of Genes and Genomes (KEGG) metabolic pathways that characterized each cluster: (**a**) linear discriminant analysis (LDA) scores for identified metabolic pathways of the *Prevotella*-rich cluster (20 pathways), the *Faecalibacterium*-rich cluster (11 pathways), and the *Bifidobacterium*-rich cluster (16 pathways). Purple indicates *Prevotella*-rich cluster, blue indicates “Other” cluster, green indicates *Faecalibacterium*-rich cluster, and red indicates *Bifidobacterium*-rich cluster; (**b**) cladogram of LEfSe analysis results. This is a circular tree diagram representing the hierarchical categories of KEGG metabolic pathways, with each yellow dot representing a pathway identified in this analysis. The regions that characterize each cluster are shaded with the color of the cluster.

**Table 1 microorganisms-10-01044-t001:** Comparisons of the patients’ background between healthy controls and quiescent ulcerative colitis patients.

	HC (*n* = 59)	qUC (*n* = 59)	*p*-Value
Sex, *n* (%)			1
Male	32 (54.2)	32 (54.2)	
Female	27 (42.8)	27 (42.8)	
Age (years)			1
Mean ± SD	50.5 ± 15.1	50.4 ± 15.5	
Median (min, max)	51.0 (24, 82)	51 (21, 84)	

HC, healthy control; qUC, quiescent ulcerative colitis; SD, standard deviation.

**Table 2 microorganisms-10-01044-t002:** The 10 most abundant genera in healthy controls and patients with quiescent ulcerative colitis.

	Healthy Controls		Patients with Quiescent Ulcerative Colitis	
	Genus	Mean	Genus	Mean
1	*Bacteroides*	17.77%	*Bacteroides*	15.71%
2	*Faecalibacterium*	8.73%	*Bifidobacterium*	14.82%
3	*Bifidobacterium*	6.55%	*Faecalibacterium*	6.58%
4	*Blautia*	5.87%	*Blautia*	6.08%
5	*Lachnospiraceae*; Other	5.15%	*Lachnospiraceae*; Other	4.32%
6	*Enterobacteriaceae*; Other	3.86%	*Roseburia*	3.67%
7	*Prevotella*	3.48%	*Enterobacteriaceae*; Other	2.98%
8	*Ruminococcaceae*; *Ruminococcus*	3.44%	*Collinsella*	2.91%
9	*Coprococcus*	2.80%	*Streptococcus*	2.91%
10	*Roseburia*	2.49%	*Ruminococcus*	2.72%

**Table 3 microorganisms-10-01044-t003:** Comparisons of the patients’ background between sustained remission and relapse patients.

	SusRem	Relapse	*p*-Value
	(*n* = 40)	(*n* = 19)	
Sex, *n* (%)			0.69
Male	21 (52.5)	11 (57.9)	
Female	19 (47.5)	8 (42.1)	
Age (years)			0.99
Mean ± SD	50.4 ± 16.2	50.3 ± 14.3	
Median (min, max)	50.5 (21, 84)	51 (32, 77)	
Body mass index			0.77
Mean ± SD	21.8 ± 3.4	22.1 ± 4.0	
Median (IQR)	21.43 (14.8, 29.6)	22.2 (16.1, 29.7)	
UC disease duration (months)			0.12
Mean ± SD	209.6 ± 21.8	156.4 ± 31.6	
Median (min, max)	178 (18, 31)	126 (31, 349)	
Observation period until relapse (days)			-
Mean ± SD	1058.9 ± 164.1	424.8 ± 49.0	
Median (min, max)	1115 (434, 1218)	357 (27, 1065)	
Extent of disease, *n* (%)			0.02
Ulcerative proctitis	10 (25.0)	0 (0.0)	
Left-sided colitis	7 (17.5)	8 (42.1)	
Panulcerative colitis	23 (57.5)	11 (57.9)	
Medication, *n* (%)			
5-Aminosalicylicacid compounds	33 (82.5)	18 (94.7)	0.19
Sulfasalazine	9 (27.3)	0 (0.0)	
Pentasa^®^	12 (36.4)	10 (55.6)	
Asacol^®^	11 (33.3)	6 (33.3)	
Rialda^®^	1 (3.0)	2 (11.1)	
Azathioprine	6 (15.0)	5 (26.3)	0.3
Infliximab	2 (5.0)	2 (10.5)	0.43
Adalimumab	1 (2.5)	0 (0.0)	0.38
Tacrolimus	1 (2.5)	1 (5.2)	0.6
Systemic steroid	0 (0.0)	0 (0.0)	-
Local steroid	1 (2.5)	1 (5.2)	0.6
Probiotics, *n* (%)			
Intake of probiotics	19 (47.5)	11 (57.9)	0.46
Biofermin	4 (21.1)	3 (27.3)	
Lac-B	6 (31.6)	2 (18.2)	
Bio-three	0 (0.0)	1 (9.1)	
Miya-BM	9 (47.4)	7 (63.6)	

SusRem, sustained remission; Relapse, relapse; SD, standard deviation; IQR, interquartile range.

**Table 4 microorganisms-10-01044-t004:** The 10 most abundant genera in each cluster.

	*Prevotella*-Rich Cluster		*Faecalibacterium*-Rich Cluster		*Bifidobacterium*-Rich Cluster		Other Cluster	
	Genus	Mean	Genus	Mean	Genus	Mean	Genus	Mean
1	*Prevotella*	22.02%	*Bacteroides*	21.56%	*Bifidobacterium*	29.62%	*Bacteroides*	19.27%
2	*Faecalibacterium*	10.59%	*Faecalibacterium*	16.73%	*Bacteroides*	9.69%	*Blautia*	6.94%
3	*Bacteroides*	7.55%	*Bifidobacterium*	6.13%	*Lachnospiraceae*; Other	4.76%	*Bifidobacterium*	6.85%
4	*Blautia*	5.59%	*Blautia*	5.44%	*Faecalibacterium*	4.34%	*Faecalibacterium*	4.85%
5	*Lachnospiraceae*; Other	4.38%	*Lachnospiraceae*; Other	4.82%	*Blautia*	4.09%	*Lachnospiraceae*; Other	4.77%
6	*Bifidobacterium*	4.18%	*Roseburia*	3.66%	*Streptococcus*	4.01%	*Enterobacteriaceae*; Other	3.67%
7	*Enterobacteriaceae*; Other	2.88%	*Enterobacteriaceae*; Other	3.29%	*Collinsella*	3.55%	*Ruminococcaceae*; *Ruminococcus*	3.58%
8	*Coprococcus*	2.72%	*Coprococcus*	3.25%	*Ruminococcus*	3.17%	*Roseburia*	3.47%
9	*Ruminococcaceae*; *Ruminococcus*	2.68%	*Lachnospira*	2.88%	*Enterobacteriaceae*; Other	3.15%	*Megamonas*	2.96%
10	*Gemmiger*	2.66%	*Ruminococcaceae*; *Ruminococcus*	2.24%	*Coprococcus*	2.49%	*Ruminococcus*	2.77%

## Data Availability

SRA records of metagenomic data of this study can be accessible with the following link: https://www.ncbi.nlm.nih.gov/sra/PRJNA804422, accessed on 8 February 2020. All data generated or analyzed during this study are included in this article. Further inquiries can be directed to the corresponding author.

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
