# Peer review of "Gut Microbiota Associated with Clinical Relapse in Patients with Quiescent Ulcerative Colitis"

_microorganisms, 2022, doi:10.3390/microorganisms10051044_

Round 1

Reviewer 1 Report

The authors evaluated the fecal microbiota of Japanese patients with quiescent ulcerative colitis (qUC) and relapse-associated microbiota, comparing data from 59 qUC patients and 59 controls. The study is relevant, well-written, and appears to have been well-conducted. I have only a minor suggestion for the discussion section:

- I suggest that the authors further discuss the clinical implications of the study results. For example, the authors could relate their findings to the potentialities of fecal transplantation, or discuss further how dietary modifications can help manipulate the gut microbiota then and benefit qUC patients.

Author Response

Dear reviewer 1

Thank you for your gentle review. Please see the attachment.

Reviewer 2 Report

The article focused on “Gut microbiota associated with clinical relapse in patients with quiescent ulcerative colitis”. The topic is of clinical importance and interesting. There were some suggestions:

  1. In Introduction part. Line 45-46 “there are conflicting reports regarding…..”. Please clarify in detail what is the conflicting reports?
  2. Line 57 “single studies are unable to capture the broad clinical spectrum of UC….” Please clarify this sentence. And is this study not a single study?
  3. Line 59-61. “Additionally, there are no studies to date on fecal microbiota involved in the clinical relapse in patients with qUC. Therefore, in this study, we analyzed the fecal microbiota of patients with UC in clinical remission and compared them with healthy controls (HCs).” The rationale is very strange. To analyze the microbiota involved in the clinical relapse in patients with qUC, the control group should be UC without replase.
  4. In Method part, line 74-75 “Patients with UC in clinical remission were prospectively selected from….” Please describe the inclusion and exclusion criteria. And how to select these patients to avoid selection bias?
  5. Line 77 “A total of 59 age- and sex-matched volunteers were selected from our previous study” Please describe the detail of the selection process and how to avoid selection bias?
  6. Please describe in Method part. Does patients with previous known gastrointestinal infectious disease, such as C. difficile infection, Salmonella or norovirus infection excluded from this study?
  7. Please describe in Method part. How to avoid the impact of antibiotics exposure on the microbiota of these patients with UC during the study period? Since health volunteers avoid antibiotics exposure, the impact of antibiotics on microbiota would be the confounding factor.
  8. In Result please describe how many patients were excluded due to antibiotics use or develop disease during the study period?
  9. In Result part. In reality there were no obvious difference in microbiota between patients with UC with or without relapse. However there were many confound factors such as drug use or probiotics use. Suggest to analyze the difference in microbiota after adjusting these confounding factors.
  10. In Discussion part. Line 293-294. “Prevotella, Faecalibacterium, and Bifidobacterium as the gut microbial genera associated with relapse.” If the control group is the health group instead of UC without relapse, it is too arbitrary to have this conclusion.
  11. In conclusion UC without relapse should be used as control group instead of health group in this study.

Author Response

Dear reviewer 2

Thank you for your gentle review. Please see the attachment.

Reviewer 3 Report

The paper entitled “Gut microbiota associated with clinical relapse in patients with quiescent ulcerative colitis” analysis bacterial microbiome differences in a cohort of ulcerative colitis patients and healthy people, and also differences in relapse and quiescent individuals. The information presented is of great interest due to the implications in disease study and clinical evolution monitorization. Some major and minor comments are asked before being accepted for publication.

 Major comments

It would be of interest to include, not only the differentiators but also the profile found (a short characterization with the 10-20 most abundant taxa) in UC/qUC.

Had  the authors physiological parameters? Correlation assays taxa/disease parameters (intake of probiotics also) with MaAslin2 or other approaches would improve the quality of the paper

L252-263: What is the profile for  the fourth “others” group? This group include the highest number of individuals (61), almost 50% UC. Do these individuals do not have any of the taxa? What are the criteria for the inclusion in this group and in the others? Related to that, did the authors to identify groups of gut microbiota communities at genus level using Dirichlet multinomial mixture by Laplace or similar to check the distribution and clustering within this population?

As PICRUST is done, authors could use this data to analyze differences in HC vs UC including all the individuals, and SusRem and Relapse vs HC

L322-323 “Although we have no data on dietary elements, it is possible to speculate, based

on the abundance of Prevotella, that the relapse group may have had a low-fiber intake and the control group had a high-fiber diet or a more balanced diet.” It is difficult (nearly not possible) to extrapolate any speculation in diet. It would be convenient to avoid it and deleted from the document.

In discussion, out of the 20 pathways “activated” in Prevotella group, only one is cited and discussed. Could authors include a deeper discussion on the other? (at least the most highlighted)?  

I suppose that there were not differences regarding butyrate pathways in PICrust analysis. Also, Song and coworkers [J. Allergy Clin. Immunol. 2016, 137, 852–860] reported a model in which atopic dermatitis was associated with an imbalance in F. prausnitzii non-butyrate producer strains. So that, differences in this taxa odes not directly imply butyrate changes. Some studies also correlate Faecalibacterium with severity in gut inflmatory profiles (Climent et al 2021, Song and coworkers 2016)

In limitations, the PICRUST analysis is an extrapolation of pathways coming from genome information. Strain-specific differences will not be considered in this case, so robustness is depending on taxonomical conserved pathways. As authors well explained, as no nutritional neither metabolomic characterization was done, the functional analysis needs to be considered as a “draft”. Please include a limitation in this way.

Minor comments. In abstract, please use italics in Genus names.

L53: Please change to “gut bacterial composition” as all these papers cited only analyzed bacteria, none of them fungi or virus.

L59: maybe include “Additionally,  to our knowledge, there are no studies to date on”

Table1 . can be adjusted for better visualization?

L117.Please recheck if it was MiSeq Reagent v3 or MiSeq Reagent v2, I think 2x250 is in v2 kit, and v3 has 2x300

Figure 1 and Figure 2, Figure 4 have very low resolution and quality. Please increase their quality. LDA score is of high interest, but it would be also interesting to include in the figures the data about the abundance of each (in the figure, or within the document)

L307-308  “When analyzed by the presence or absence of probiotics, the tendency of the results for these three genera was similar (data not shown).” Could authors include the study in the results? It is important to include not only the significant data but also the trend and not significant, to discard correlations.  Also please include the groups compared, the statistics, the software used and if other parameters were also analysed and resulted non significant.

MY apologizes if there is a misunderstanding, in the L85 it states “patients with serious metabolic, respiratory, cardiologic, renal, hepatic, hematologic, neurologic, or psychiatric functions and those who regularly used prebiotic or probiotic medications were excluded” but in Table 2, a high percentage of individuals consumed probiotics. Is it ok? Is there any information missing?

L388 “They may be characterized by enterobacterial genera other” The fourth group was defined by the authors, who would know if it is characterized or not by enterobacterial genera. Is it “may” ok?

L404 please change “We confirmed the difference in the microbiota between qUC and HC in our study, and …”

Author Response

Dear reviewer 3

Thank you for your gentle review. Please see the attachment.

Round 2

Reviewer 2 Report

had been revised as suggestions